# Mutational Diversity in the Quinolone Resistance-Determining Regions of Type-II Topoisomerases of *Salmonella* Serovars

**DOI:** 10.3390/antibiotics10121455

**Published:** 2021-11-26

**Authors:** Aqsa Shaheen, Anam Tariq, Mazhar Iqbal, Osman Mirza, Abdul Haque, Thomas Walz, Moazur Rahman

**Affiliations:** 1Department of Biochemistry and Biotechnology, Hafiz Hayat Campus, University of Gujrat, Gujrat 50700, Pakistan; 2Drug Discovery and Structural Biology Group, Health Biotechnology Division, National Institute for Biotechnology and Genetic Engineering (NIBGE), Faisalabad 38000, Pakistan; anamtariq.biotech@gmail.com (A.T.); mazhar@nibge.org (M.I.); 3Department of Drug Design and Pharmacology, Faculty of Health and Medical Sciences, University of Copenhagen, 2200 Copenhagen, Denmark; om@sund.ku.dk; 4Human Infectious Diseases Group, Akhuwat First University, Faisalabad 38000, Pakistan; haque.hameed@akhuwat.org.pk; 5Laboratory of Molecular Electron Microscopy, Rockefeller University, New York, NY 10065, USA; twalz@rockefeller.edu; 6School of Biological Sciences, Quaid-I-Azam Campus, University of the Punjab, Lahore 54590, Pakistan

**Keywords:** *Salmonella*, fluoroquinolones, quinolone resistance-determining regions (QRDRs), food-borne pathogens, plasmid-mediated quinolone resistance (PMQR), DNA gyrase, topoisomerase IV, typhoidal *Salmonella*, non-typhoidal *Salmonella* (NTS)

## Abstract

Quinolone resistance in bacterial pathogens has primarily been associated with mutations in the quinolone resistance-determining regions (QRDRs) of bacterial type-II topoisomerases, which are DNA gyrase and topoisomerase IV. Depending on the position and type of the mutation (s) in the QRDRs, bacteria either become partially or completely resistant to quinolone. QRDR mutations have been identified and characterized in *Salmonella enterica* isolates from around the globe, particularly during the last decade, and efforts have been made to understand the propensity of different serovars to carry such mutations. Because there is currently no thorough analysis of the available literature on QRDR mutations in different *Salmonella* serovars, this review aims to provide a comprehensive picture of the mutational diversity in QRDRs of *Salmonella* serovars, summarizing the literature related to both typhoidal and non-typhoidal *Salmonella* serovars with a special emphasis on recent findings. This review will also discuss plasmid-mediated quinolone-resistance determinants with respect to their additive or synergistic contributions with QRDR mutations in imparting elevated quinolone resistance. Finally, the review will assess the contribution of membrane transporter-mediated quinolone efflux to quinolone resistance in strains carrying QRDR mutations. This information should be helpful to guide the routine surveillance of foodborne *Salmonella* serovars, especially with respect to their spread across countries, as well as to improve laboratory diagnosis of quinolone-resistant *Salmonella* strains.

## 1. Introduction

‘Quinolones’ are a class of bactericides that include all synthetic drugs containing a quinolone or naphthyridone nucleus [1]. Quinolones exert their antibacterial effect by inhibiting bacterial DNA synthesis. The first synthetic quinolone, nalidixic acid, was introduced in 1962 to treat urinary tract infections [2]. Nalidixic acid and related first-generation antibiotics were only active against Gram-negative bacteria. Efforts to develop broad-spectrum antibacterial drugs by modifying nalidixic acid resulted in the introduction of “fluoroquinolones” (FQs) in 1980, which feature a fluorine atom at position C-6 of the quinolone ring [2] (Figure 1). While norfloxacin was the first FQ, used to treat urinary tract infections, ciprofloxacin, which was introduced in the United States in 1987, was widely prescribed to also treat various other infections. Ciprofloxacin was found to be very effective against Gram-negative bacilli, but it was less active against Gram-positive bacteria [3]. In 1991, another FQ analogue was introduced, ofloxacin, which was particularly effective in treating systemic infections [4].

It should be noted that not all cases of salmonellosis are treated with quinolones. The reason is that it is generally not recommended to use antibiotics to treat non-typhoidal *Salmonella* (NTS) gastroenteritis, because it is usually self-limiting [5]. However, FQs are prescribed if *Salmonella* gastroenteritis is anticipated to aggravate the other disorders of a patient, such as cardiovascular diseases, immunosuppressed conditions, and malignant diseases. Treatment with FQ is also recommended when patients show symptoms of severe sepsis, such as high body temperature, impaired kidney function, and systemic toxicity [5]. FQs, particularly ciprofloxacin and ofloxacin, are excellent drugs to cure typhoidal and non-typhoidal salmonellosis as well as to clear asymptomatic bacillus carriers of the bacteria [6,7,8].

However, widespread use of FQs has resulted in the appearance of chromosomal and plasmid-mediated resistance mechanisms. This review focuses on FQ resistance caused by target-site mutations, i.e., mutations in the quinolone resistance-determining regions (QRDRs; see below) of type-II topoisomerases.

## 2. Mode of Action of Fluoroquinolones and QRDRs

The bactericidal mode of action of FQs is through the inhibition of the bacterial DNA-synthesis machinery, specifically by interfering with the type-II topoisomerases DNA gyrase and topoisomerase IV [1]. The physiological role of DNA gyrase is to introduce negative supercoiling in DNA [9], so that a DNA molecule, which can be 1000 to 10,000 µm long, can be sufficiently compacted to fit inside the limited volume of a bacterium [10]. The functional unit of bacterial DNA gyrases is a tetramer, consisting of two GyrA and two GyrB subunits [11]. The GyrA subunit is responsible for breaking and resealing the target DNA strand and the GyrB subunit provides the required energy through ATP hydrolysis [11]. Topoisomerase IV, which is critical for proper daughter chromosome separation during replication, is also a tetramer and consists of two ParC and two ParE subunits [10,11,12]. During replication, the ParC subunit co-localizes with the replisome [12], which requires a ParE subunit with an active catalytic ATPase site [13]. The GyrA subunit of DNA gyrase is homologous to the ParC subunit of topoisomerase IV, and the GyrB subunit to the ParE subunit [14].

FQ antibiotics inhibit type-II topoisomerases after they bind to DNA and form a transient complex, in which the enzyme’s active-site tyrosine (Tyr) residue forms a covalent bond with a DNA phosphate ester, thus hindering the resealing process and resulting in the accumulation of double-stranded DNA fragments [15,16]. DNA gyrase is usually referred to as the primary target of quinolones in Gram-negative pathogens such as *E. coli* and *Salmonella* [13,17], whereas topoisomerase IV is considered to be the primary target of quinolones in Gram-positive bacteria such as *Streptococcus*. However, there are many exceptions to this rule. For example, in some Gram-positive species, FQs target both DNA gyrase and topisomerase IV [18,19], and in other species, different quinolones have distinct primary targets [16].

Quinolone resistance is centered on a small stretch of amino acid residues, commonly referred to as the quinolone resistance-determining regions (QRDRs), of bacterial DNA gyrase and topoisomerase IV. These QRDRs are hotspots for mutations that result in quinolone resistance.

Initially, Yoshida et al. defined the QRDR of GyrA from the alanine (Ala) at position 67 to the glutamine (Gln) at position 106 (numbering for the *E. coli* protein) [20] (Figure 2A) and that of GyrB from aspartate (Asp) 426 to lysine (Lys) 447 [21] (Figure 2B). Some researchers described the QRDR of *Salmonella* GyrA to contain amino acids 67 to 122 [22]. The QRDR of ParC encompasses the residues between Ala64 and Gln103 (Figure 2C) and that of ParE those between Asp420 and Lys441 (Figure 2D).

The Genbank accession numbers of all these genes are given in Table 1.

According to Afzal et al., QRDRs include: amino acids Asp36 to Gly151 of GyrA, Gly405 to Glu520 of GyrB, al46 to Leu133 of ParC, and Glu449 to Ile529 of ParE [24]. One should be aware, however, that further research is likely to expand the QRDR regions. In fact, the QRDR of GyrA has already been proposed to include the residue at position 51, because of its critical role in quinolone resistance [25]. Similarly, the proposed involvement of arginine (Arg) 121 in the formation of a salt bridge when GyrA interacts with ciprofloxacin [26] prompted some researchers to expand its QRDR to residues downstream of Gln106 [22,24]. Likewise, complete sequencing of the *gyrA* gene of quinolone-resistant isolates has implicated mutations outside the QRDR in FQ resistance [27].

In the last ten years, a growing number of publications addressed the issue of quinolone resistance among clinical and veterinary isolates of *Salmonella* [28,29,30,31]. However, there is currently no comprehensive overview that describes the diversity of mutations in QRDRs of different *Salmonella* serovars and their relationship with plasmid-mediated quinolone resistance (PMQR) determinants. This review aims to fill this gap and to collate available information, with a focus on recent findings, to provide a snapshot of current QRDR mutations and their relevance to circulating strains of different *Salmonella* serovars.

## 3. Known Mechanisms of Quinolone Resistance

Selection pressures imposed by antibiotics such as quinolones and the evolutionary principle of survival of the fittest have led pathogens, including *Salmonella*, to develop a variety of evasion mechanisms [32,33,34]. *Salmonella* express chromosomally and plasmid-encoded proteins that protect them against quinolones. In terms of chromosomally encoded proteins, mutations can alter the drug targets [35,36,37] or affect local and global repressors that result in down-regulated expression of outer membrane porins [38,39] or up-regulated expression of inner membrane efflux transporters [40,41,42,43]. These defense mechanisms can be further enhanced by transferable resistance elements residing on plasmids.

Plasmid-borne quinolone-resistance determinants work through three distinct mechanisms. First, Qnr proteins [44], a family of pentapeptide repeat proteins [45,46], interfere with the action of FQs on type-II topoisomerases either by reducing the formation of replication forks, thus reducing the recruitment of topoisomerases to DNA, or, if the replication fork has already formed, by occupying the sites in the replication fork where quinolones would bind [47]. Based on their sequence similarity, Qnr proteins have been further divided into the qnrA [44], qnrB [48], qnrC [49], qnrD [33], QnrS [50] and QnrCV [51] families [13]. Second, the aminoglycoside-(6)-N-acetyltransferase encoded by the *aac(6′)-Ib-*cr gene, which differs from that encoded by the *aac(6′)-Ib* gene by the presence of two point mutations, Trp102Arg and Asp179Tyr, N-acetylates and thus inactivates quinolones [52]. Third, efflux transporters, such as OqxAB [53], QepA1 [54,55], QepA2 [56] and QepA3 [57], contribute to quinolone resistance by extruding antibiotics from the bacteria.

The relative capacity of the various mechanisms to confer FQ resistance varies. Ciprofloxacin is usually chosen as a representative FQ in minimum inhibitory concentration (MIC) assays to evaluate the resistance level of clinical isolates. Clinical and laboratory standard institute (CLSI) guidelines [58] are followed to ensure a uniform classification of isolates showing ciprofloxacin resistance. According to these guidelines, resistant isolates have an MIC of 1–2 mg/L, isolates with intermediate susceptibility have an MIC of 0.12–0.5 mg/L, and susceptible isolates have an MIC of ≤0.06 mg/L. Measured in this way, chromosomally encoded efflux proteins increase the MIC of ciprofloxacin by a factor of 2 to 8 [59,60,61], and porin downregulation increases the MIC by a similar factor [62], whereas mutations in topoisomerases account for a 10- to 60-fold increase in the MIC, depending on the number and position of the mutations [63,64]. For plasmid-encoded proteins, the efflux pumps QepA and OqxAB increase the MIC of ciprofloxacin by a factor of 10 to 16 [53,55,65], Qnr proteins account for a 16- to 125-fold increase in the MIC [66], and the quinolone N-acetylating enzyme is responsible for a 4-fold increase in the MIC [52].

## 4. Mutations in the QRDRs of *Salmonella* Serovars

*Salmonella* is a diverse genus, primarily divided into two species, *S. enterica* and *S. bongori*. *S. enterica* is further divided into six subspecies. *S. enterica* subsp. *enterica* (>1500 members) is the largest subspecies [67] and is of primary interest for public health. Members of *S. enterica* subsp. *enterica* have adapted to warm-blooded vertebrates (birds and mammals) [68] and these *Salmonella* serovars are the leading food-borne pathogens. Ciprofloxacin resistance in food-borne *Salmonella* has been reported worldwide [69]. *Salmonella* Enteritidis and *Salmonella* Typhimurium are the isolates identified in food that are most commonly implicated as the source of salmonellosis, and there have been numerous reports on mutations in the QRDRs of isolates. Other serovars are usually limited to certain geographical regions and are therefore less well surveilled. However, global travel and food import have substantially increased the risk of salmonellosis associated with less prevalent serovars. The current review covers the available information related to the *Salmonella* serovars listed in Table 2.

### 4.1. Mutations in the QRDRs of Typhoidal Salmonella

Typhoidal *Salmonella*, including *S.* Typhi and *S.* Paratyphi A–C, are a major cause of morbidity and mortality in developing countries and among travelers visiting areas where these pathogens are endemic [36]. The persistence of typhoidal *Salmonella* is mostly due to poor sanitation and unhygienic conditions but food handlers carrying typhoidal *Salmonella* have also been suggested to contribute to the high endemicity of typhoid fever [70,71].

Analysis of QRDR mutations in *S.* Typhi isolates showed that reduced ciprofloxacin susceptibility in this serovar is most often caused by a substitution of Ser83 to phenylalanine (Phe) in GyrA (Appendix A), a mutation that has been found worldwide [24,64,72,73,74,75,76,77,78,79,80] (Figure 3 and Figure 4). Analysis of the literature from Asia, in particular from Southeast Asia, revealed that in India, Bangladesh, Pakistan and Nepal, the Ser83Phe substitution is the most prevalent mutation in ciprofloxacin-resistant *S.* Typhi isolates, with an MIC ranging from 0.9 to 3 mg/L, and the Ser83Tyr substitution is the second most frequent mutation [24,64]. Most of these studies reported the absence of PMQR determinants. While it had been proposed earlier that Ser83 is an important site for imparting partial FQ resistance in *S.* Typhi and *S.* Paratyphi A [81], complete ciprofloxacin resistance requires double mutations in the QRDR of GyrA [35]. Complete ciprofloxacin resistance of *S.* Typhi in the absence of a double mutation in the *gyrA* suggests that these isolates also acquired an active efflux transporter or have impaired outer membrane porins. This has been observed in an *S.* Typhi isolate from South Africa with an MIC of 4 mg/L, which carried only a single substitution in a *gyrA* gene, namely Ser83Tyr, and the presence of *qnrS1*, but also featured an efflux system that contributed to its ciprofloxacin resistance [82]. However, most of the studies examining ciprofloxacin resistance in *S.* Typhi have not investigated the contribution of all the possible resistance mechanisms (Appendix A).

Of note, *S.* Typhi H58 strains generally seem to carry the Ser83Phe mutation in GyrA [83]. This strain is prevalent in Vietnam, India, Bangladesh, Pakistan, Laos and Nepal [83] and cases reported in China, France, Kenya and Japan can be linked to travel to these areas [72,75,84,85].

Another residue that has been found to be mutated in most studies is Asp at position 87. It has been found to be mutated into Asn, Gly, Tyr and Val [72,74,86] (Figure 3). Two independent studies reported ciprofloxacin-resistant isolates with two substitutions in GyrA, Asp87Val and Ser83Phe [34,86]. The Asp87Val substitution is rare and imparts resistance (Appendix A). Koirala et al. were the first to describe this mutation and a comparison with already known double mutants, namely Ser83Phe/Asp87Asn and Ser83Phe/Asp87Gly, established that it was the Asp87Val mutation that was responsible for the increased ofloxacin resistance [86]. Another resistance-conferring substitution in GyrA, Asp87Tyr, is not commonly found in *S.* Typhi and has only been reported for isolates from Belgium [34] (Appendix A). An additional uncommon mutation in *S.* Typhi is Ser83Leu [34,87]. A novel mutation reported for an isolate from Italy, Asp82Asn, was found not to confer resistance to ciprofloxacin [76].

In addition to the most frequently mutated GyrA residues in typhoidal *Salmonella*, Ser83 and Asp87, the Glu133Gly substitution, first identified only in *S.* Typhi and later also in *S.* Typhimurium, has also been suggested to confer ciprofloxacin resistance (Appendix A). However, the wild-type residue at position 133 is controversial for *S.* Typhi, because an analysis of available GyrA sequences in GenBank shows that a Gly is more common in this position [88], and Ceyssens et al. reported that 50% of Belgian *S.* Typhi isolates also have a Gly at this position [34]. Furthermore, Eibach et al. found that an *S.* Typhi isolate carrying only the Glu133Gly mutation was susceptible to ciprofloxacin with an MIC of 0.06 mg/L [89], implying that a Gly at this position does not confer resistance. Similar findings were made for isolates from Cambodia [90]. Another study on *S.* Typhi isolates collected in Iran reported that Ser83Phe/Gly133Glu double mutants were more susceptible to nalidixic acid than single Ser83Phe mutants [91]. An earlier study found an isolate from India with the identical double mutation in GyrA to be resistant to nalidixic acid [92], but this discrepancy was later resolved by the identification of an efflux activity conferring resistance to this strain [91].

Mutations in GyrB occur far less frequently in *S.* Typhi, but occur for Gly435 [76], Ser464 [72,93], Gln465 [72], Glu466 [72], Ala468 [72] and Ala574 [34] (Appendix A) (Figure 5). The Ser464Phe mutation is particularly important for conferring a non-classical resistance phenotype, in which strains are susceptible to nalidixic acid but display an intermediary susceptibility to ciprofloxacin [72]. The Gln465Leu, Glu466Asp, and Ala468Glu mutations do not confer ciprofloxacin resistance [72]. Mutations in ParC are even less common, with the preferred sites being Thr57, which is almost always mutated to Ser, and Ser80, which can be substituted by an Arg or Ile, with the latter mutation being found in *S.* Typhi (Appendix A). Eaves et al. suggested that the Thr57Ser mutation reduces the binding affinity for quinolone, as a result of the smaller Ser side chain [94], but Qian et al. described it as a compensatory mutation that actually caused the *Salmonella* isolates to become susceptible to ciprofloxacin [95]. Other uncommon ParC mutations in *S.* Typhi include Glu92Lys [85] and Trp106Gly [96]. The presence of a Glu residue at position 92 needs to be further investigated, because our current survey found this position to be usually occupied by a proline (Figure 6). Mutations in ParE were reported for isolates from France and Italy that were collected from travelers returning from India. The Asp420Asn mutation appears to increase ciprofloxacin susceptibility [72] (Figure 7), isolates carrying only a Ser83Phe substitution are more resistant to ciprofloxacin, and the effect of the Ser493Phe mutation in ParE awaits further investigation [76].

Mutations in *S.* Paratyphi strains are not well documented. Among the three strains causing paratyphoid fever, *S.* Paratyphi A is more prevalent and is therefore comparatively better studied in terms of epidemiology and resistance mechanisms. In 2010, a large-scale outbreak of paratyphoid fever was reported in China [97], and in 2013, another outbreak occurred among Japanese travelers returning from Cambodia [98]. An investigation using whole-genome sequencing revealed that a strain carrying the Ser83Phe mutation in GyrA originated in Cambodia and was then spread by travelers into other countries [71,97]. The Ser83Tyr and Asp87Asn mutations were the second and third most abundant mutations found in this study but occurred significantly less frequently than Ser83Phe [71]. *S.* Paratyphi A, B and C strains with mutations in their QRDRs have been reported for isolates from Europe [34], Africa [99] and Asia [91,95,100]. With only few exceptions, all these isolates carried a single mutation in GyrA either at position 83 or 87 (Appendix A). Double mutations were reported for Iranian isolates with *S.* Paratyphi B carrying Arg47Ser and Asp87Gly and *S.* Paratyphi C carrying Asp147Gly and Ser83 Phe [91]. Arg47Ser and Asp147Gly are novel mutations and their contribution to conferring ciprofloxacin resistance is not yet clear. One of the *S.* Paratyphi A isolates with high ciprofloxacin resistance, i.e., an MIC of 32 mg/L, has a double mutation in GyrA, Ser83Phe and Asp87Gly [100]. This isolate was found to be negative for the presence of PMQR genes but the contribution of efflux was not investigated. An earlier study of an *S.* Paratyphi A isolate from Japan showed an unusually high ciprofloxacin resistance with an MIC of 128 mg/L. The basis of this very high resistance was identified as double mutations in GyrA, Ser83Phe and Asp87Asn, together with a Glu84Lys mutation in parC [101]. Recently, *S*. Typhi and *S*. Paratyphi B isolates from Jiangsu Province of China have been reported to carry many novel mutations in GyrB, ParC and ParE (Appendix A) [95]. However, their relevance for ciprofloxacin resistance awaits further study. Moreover, the nature of the wild-type residue at position 498 requires further analysis, our sequence alignment of a representative typhoidal *Salmonella* strain, i.e., *S*. Typhi str. Ty2, with a representative NTS strain, i.e., *S*. Typhimurium str. LT2 and *E. coli* O157:H7 str. EDL933 suggests that the wild-type residue at this position is Ile (Figure 7), and not Ser as has been stated before [95].

The diverse mutations found in the QRDRs of typhoidal *Salmonella* are summarized in Figure 3. These sites of mutations for *gyrA*, *gyrB*, *parC* and *parE* are also explained in the aligned sequences in Figure 4, Figure 5, Figure 6 and Figure 7, respectively.

The PMQR genes detected in *S.* Typhi isolates carrying QRDRs mutations are *qnrS1* and *qnrB2* (Appendix A). The *qnrS1* gene has been reported in isolates from different regions [82,85,102], but its origin could be traced back to Southeast Asia. It should be noted that quinolone efflux is rarely investigated and this resistance mechanism is thus likely underappreciated. According to Adachi et al., the different MICs of isolates with the same QRDR mutation can be explained by the involvement of other quinolone-resistance mechanisms [101], for example, for *S.* Typhi isolates with the Ser83Phe substitution that show MICs from 0.19 to 3 mg/L [24,34,72,75,100] (Appendix A). Taken together, *S.* Typhi and *S.* Paratyphi A appear to respond to ciprofloxacin pressure predominantly by adopting the Ser83Phe mutation in GyrA. Since *S.* Paratyphi B and C are less prevalent, large-scale studies in endemic areas are needed to evaluate their serovar-specific mutation resulting from antibiotic pressure.

### 4.2. Mutations in the QRDRs of Non-Typhoidal Salmonella

Non-typhoidal *Salmonella* (NTS) are most often contracted through the consumption of contaminated food and water. While the thermal processing of food is considered to be an effective method to eliminate NTS, NTS infections caused by the consumption of thermally treated food are still reported around the globe. The predominant serovars of foodborne *Salmonella* are *S.* Typhimurium, *S.* Enteritidis, *S.* Newport, *S.* Indiana, *S.* Kentucky, and *S.* Virchow [103,104]. There have been many studies on mutations in the QRDRs of *S.* Enteritidis and *S.* Typhimurium, because of their universal prevalence and because they cause most gastroenteritis-associated infections. Other *Salmonella* serovars are less often associated with gastroenteritis and so there are less studies on their QRDR mutations.

#### 4.2.1. *Salmonella* Enteritidis (*S*. Enteritidis)

*S.* Enteritidis is the prime cause for egg-borne NTS infections in developed countries [29,105]. A study by Lunn et al., on clinical isolates of *S.* Enteritidis in Barcelona, found that the Asp87Tyr substitution in GyrA is the predominant mutation in *S.* Enteritidis isolates [106]. Another study from Spain also found this mutation in the majority of *S.* Enteritidis isolates, which exhibited MICs of ciprofloxacin ranging from 0.03 to 2 mg/L [107]. Our literature review finds that Asp87Tyr is also the predominant substitution in *S.* Enteritidis isolates from Malaysia [108,109], Korea [110,111], Africa [89], Brazil [112,113] Belgium [34], Japan [114] and Thailand [115] (Appendix A).

The Asp87Tyr substitution is dominant in isolates from different origins, but substitutions to Gly and Asn are found at a lower frequency [89,91,115,116,117]. Isolates from Iran and Serbia showed predominantly substitutions of Asp87 with Asn and Gly/Asn, respectively, but no substitutions with Tyr [91,116]. Other substitutions at position 87 are rare and include one to Ser in an Egyptian isolate [39], which will be discussed in more detail below.

Ser83 of GyrA is mostly substituted with Phe in *S.* Enteritidis isolates with reduced or complete ciprofloxacin resistance, but substitution to Tyr has also been reported and has been implicated in reduced susceptibility to ciprofloxacin. A unique Ser83Ile substitution has been reported for a Malaysian isolate [108] (Appendix A), and this mutation has recently been shown to confer a high level of quinolone resistance [118]. A novel mutation in GyrA, Ser97Pro, was found in an *S.* Enteritidis isolate from Brazil, but it does not seem to contribute to quinolone resistance [113] (Appendix A). Double mutations in GyrA are not common in *S.* Enteritidis and have only been reported for a few isolates from Belgium [34], Brazil [112] and Thailand [115,119]. No mutations have been found in the GyrB subunit in any isolates analyzed to date. Four mutations in topoisomerase IV subunit ParC, Thr57Ser [32], Ser67Cys, Arg76Cys, Cys80Arg [39] (discussed later in detail) and one mutation in topoisomerase IV subunit ParE, Val521Phe [108] (Appendix A) have been reported.

A clinical *S.* Enteritidis isolate was found to have a double mutation in GyrA, Ser83Phe and Asp87Ser (a rare substitution at this position), a novel triple mutation in ParC, Ser67Cys, Arg76Cys and Cys80Arg (Appendix A), as well as an active efflux system, resulting in a very high MIC of ciprofloxacin of 256 mg/L. Inhibition of the efflux system reduced the MIC of this isolate by a factor of 2, 3 or 4, depending on whether norepinephrine, carbonyl cyanide m-chlorophenylhydrazone or trimethoprim was used as an inhibitor [39]. It is worth mentioning that these inhibitors work in synergy with ciprofloxacin. This study by Rushdy et al. also shed light on the role of the altered expression of outer membrane porins in conferring high resistance to ciprofloxacin, but it did not explore the possible involvement of plasmid-mediated resistance. Overall, this study explained the role of some underrepresented quinolone-resistance mechanisms in greater depth. On the other hand, the novel mutations need further investigation, because the study based its analysis on an outdated nucleotide sequence that has been replaced with a sequence of *S*. Typhimurium str. LT2 (Genbank accession number AE006468). Our sequence alignment for ParC (Figure 6) shows that the wild-type residue for *S*. Typhimurium str. LT2 at position 67 and 76 are Val and Pro, respectively, rather than Ser and Arg as stated in the study. Similarly, as has been reported in many studies surveyed in this review, the wild-type residue at position 80 is not a Cys.

The importance of GyrA mutations at positions 83 and 87 in mediating ciprofloxacin resistance has been documented by many studies. It can also be speculated that the rare substitutions of residue at position 87 confer resistance because of its critical position in the enzyme. On the other hand, due to uncertainty regarding the nature of the wild-type residues, it is not possible to assess the capacity of the ParC mutations described above in conferring a resistance phenotype to *Salmonella* strains [39].

With respect to PMQR-determining elements in isolates with QRDR mutations, *S.* Enteritidis harbors the *qnrS1* gene in isolates from Brazil, Malaysia, Belgium and Thailand, whereas the *qnrB* and *aac(6′)-lb-cr* genes have only been reported in isolates from Brazil (Appendix A). Another study from China reported that PMQR determinants, i.e., the *qnr* and *aac(6′)-Ib-cr* genes, are frequently detected in *S.* Enteritidis as well as in *S.* Derby, *S.* Agona and *S.* Typhimurium [103].

Some *S.* Enteritidis isolates also exhibit the non-classical phenotype, i.e., complete susceptibility to nalidixic acid and reduced susceptibility to ciprofloxacin with an MIC of ciprofloxacin in the range of 0.125–1 mg/L [107]. This phenotype in *S.* Enteritidis has been suggested to not be the result of the GyrB mutation observed for the typhoidal isolates described above, but instead to be due to the presence of PMQR determinants [107]. Concern has been raised about the widespread distribution of such non-classical *Salmonella* strains, because they are not detected in routine screenings in clinical settings that use the nalidixic acid resistance assay [107,120] and because their unnoticed spread would compromise the use of ciprofloxacin for the treatment of salmonellosis.

#### 4.2.2. *Salmonella* Typhimurium (*S*. Typhimurium)

*S.* Typhimurium is the leading cause of NTS infections [121]. *S.* Typhimurium-induced acute gastroenteritis is usually linked to the consumption of contaminated pig and bovine products [104,107]. *S.* Typhimurium phage type DT104 has been found to be multidrug resistant and is widely distributed, especially in Europe and North America [122,123].

The analysis of mutations in the QRDRs of *S.* Typhimurium isolates from around the world revealed the prevalence of a single mutation in GyrA either at position 83 or 87 (Appendix A). Double mutants have been reported in isolates from China, and rare incidents reported from Belgium might be associated with travelers returning from China. Isolates carrying double mutations were also found to have an additional mutation in ParC, Ser80Arg. This ParC mutation was found in all double mutants except for one [34], which is also unusual in that it carries the Glu133Gly mutation in GyrA described above that is characteristic for typhoidal *Salmonella* (Appendix A). Some isolates with a double mutation in GyrA were found to have the Ser80Ile substitution identified in Belgium [34]. Isolates with double mutations in GyrA and a single mutation in ParC also all have the Ser83Phe substitution in common [34,124,125]. In *S*. Typhimurium isolates, Asp87 is most commonly replaced by Asn (Appendix A). In a study carried out on *S.* Typhimurium isolate from the US, the complete sequencing of *gyrA* gene instead of only QRDR region featured a single mutation in GyrA that results in the mutation of the proline (Pro) residue at position 864 to serine (Ser), and this mutation has been described to be likely associated with FQ resistance [27].

An *S.* Typhimurium isolate with exceptionally high resistance against ciprofloxacin, i.e., with an MIC of 512 mg/L, has been identified in the Middle East and is characterized by a triple mutation in GyrA, Ser83Phe, Asp87Gly and Ala119Ser, and no mutation in ParC [39]. Mutations in GyrB would be a likely possibility for the high level of resistance, but GyrB was not examined for mutations in this study. However, the study found that this *S.* Typhimurium strain did not express OmpA, OmpC and OmpF and only little OmpD [39]. Efflux inhibition with carbonyl cyanide m-chlorophenylhydrazone, trimethoprim, and norepinephrine resulted in reductions in the MIC by factors of 2, 3, and 4, respectively. Thus, while target alteration, downregulated influx and upregulated efflux all contribute to the high ciprofloxacin resistance of this isolate, it is possible that PMQR element (s) further contributed to the high resistance.

The GyrB and ParE subunits are rarely mutated in *S.* Typhimurium isolates. The only mutation in GyrB reported to date is a Glu466Asp substitution found in an isolate from Africa [89,126]. This mutation had previously been identified in typhoidal *Salmonella* isolates. Earlier studies associated another GyrB mutation, Ser464Phe, with high ciprofloxacin resistance, with MICs in the range of 16–32 mg/L [127]. The only mutation identified in ParE to date, Met438Ile, was found in an isolate from Malaysia [108], but because this isolate also carries a PMQR gene, the extent by which the ParE mutation contributes to quinolone resistance is currently unclear.

An in vitro selection study of ciprofloxacin-resistant *S.* Typhimurium isolates showed that exposure of the parent *S.* Typhimurium strain (isolated from chicken, carrying a single mutation in ParC, Thr57Ser, and having an MIC of 0.03 mg/L) to ciprofloxacin (0.25 mg/L) resulted in progeny that retained the Thr57Ser mutation in ParC but also acquired the Asn87Asp mutation in GyrA, increasing its MIC to 0.5 mg/L. A subsequent increase in selection pressure by serially increasing the ciprofloxacin concentration from 2 to 4 mg/L resulted in progeny that only retained the GyrA and ParC mutations, but displayed an increased MIC of 8 mg/L [128], implying that the bacteria picked up another ciprofloxacin-resistance mechanism distinct from mutations in the target genes. A further increase in selection pressure by increasing the ciprofloxacin concentration to 64 mg/L resulted in progeny with an additional mutation in GyrB, Ser464Phe, while retaining the GyrA and ParC mutations. The MIC of this triple mutant was 32 mg/L [128]. This study concluded that efflux mediated by the AcrAB-TolC efflux pump played a crucial role in the increase in ciprofloxacin resistance before the acquisition of the GyrB mutation. Enhanced AcrAB-TolC expression was also found in an earlier in vitro selection study with the ciprofloxacin-resistant *S.* Typhimurium strain S21 (MIC < 0.15 mg/L). Selection pressure due to ciprofloxacin concentrations ranging from 0.125 to 256 mg/L resulted in progeny that sequentially acquired first a mutation in GyrA, Ser83Phe, resulting in an MIC of 0.25 mg/L, and then a second mutation in ParC, Ser80Ile, resulting in an MIC of 4 mg/L [129]. Even though a further increase in ciprofloxacin pressure resulted only in the retention of the GyrA and ParC mutations with no additional mutations being acquired, the MIC of the isolates increased to 32 mg/L and finally to 256 mg/L. The increase in MIC without the acquisition of any additional mutations in the target genes was correlated with enhanced expression of the AcrAB-TolC efflux pump [129]. A different study on the generation of ciprofloxacin-resistant *S.* Typhimurium mutants implicated the RamA regulator in enhanced ciprofloxacin efflux [130].

Analysis of the above-mentioned in vitro-selected ciprofloxacin-resistant *S*. Typhimurium progenies and a survey of the isolates listed in Appendix A highlights the contribution of PMQR genes to quinolone resistance. Natural isolates demonstrate the PMQR determinants can enhance quinolone resistance to a level that mutations in GyrB or ParC are no longer required for survival. A clinical case reported by de Toro et al., on the in vivo selection of a quinolone-resistant *S.* Typhimurium isolate further illustrates this point. Treatment of an *S.* Typhimurium isolate, which was *qnrS1* positive and displayed reduced susceptibility to ciprofloxacin (MIC of 0.5 mg/L), with ciprofloxacin for 7 days resulted in a ciprofloxacin-resistant strain that retained the *qnrS1* gene but also acquired the *aac(6′)-Ib-cr* gene and the Ser83Tyr mutation in GyrA, elevating its MIC to 8 mg/L [131]. This level of resistance is comparable to the resistance that can be attained by additional mutations in GyrA.

PMQR determinants identified in *S.* Typhimurium isolates with QRDR mutations include the *oqxAB*, *aac(6′)-Ib-cr*, *qnrS1* and *qnrB* genes. Under laboratory settings, the horizontal transfer of the *qnrS1* gene was found to increase the MIC of ciprofloxacin by a factor of 67 [131] and the combined horizontal transfer of the *qnrS1* and *aac(6′)-Ib-cr* genes resulted in an increase in the MIC by a factor of 133 [131]. *S.* Typhimurium strains without QRDR mutations but carrying PMQR genes have also been reported to harbor the *qepA* and *qnrS1* genes [106]. A non-classical *S.* Typhmurium strain with an MIC of ciprofloxacin of 4 mg/L was found to have no QRDR mutations and no PMQR genes [107], suggesting the involvement of a different resistance mechanism, such as increased efflux mediated by inner membrane transporters or reduced influx through outer membrane porins, and possibly even some as yet unknown mechanism. Efflux mediated by the AcrAB-TolC efflux pump seems to be a characteristic feature of ciprofloxacin-resistant *S.* Typhimurium isolates, as listed in Appendix A and supported by the in vitro-generated ciprofloxacin-resistant mutants. These studies reveal that an important mechanism by which *S*. Typhimurium acquires ciprofloxacin resistance is by activating efflux pumps mediated by the RamA regulator [130].

#### 4.2.3. *Salmonella* Hadar (*S*. Hadar)

*S.* Hadar is another poultry-borne pathogen and is considered to be a reservoir of antibiotic resistance genes [132]. Our literature survey shows that *S.* serovar Hadar isolates with intermediate ciprofloxacin susceptibility from Europe and Asia mostly carry a mutation of Asp87 in GyrA (Appendix A). *S.* Hadar isolates collected from Switzerland and Poland almost all showed this single Asp87 mutation, thus suggesting that this strain spread through clonal expansion [28,133]. Thong et al. came to a similar conclusion for food-borne *S.* Hadar isolates from different locations in Malaysia. Despite their diverse origins, all isolates shared the same QRDR mutations, namely Asp87Tyr in GyrA and Thr57Ser in ParC, indicative of the clonal spread of this strain [108]. While the Asp87Tyr mutation in GyrA is dominant in *S.* Hadar isolates from Malaysia [108] and Korea [134], the Asp87Asn is more common in isolates from Belgium, Poland, Morocco and Spain [34,135]. The Asp87Gly mutation, although frequent in other *Salmonella* serovars, has not yet been reported in *S*. Hadar. Substitutions of GyrA residue Ser83 with Tyr and Phe have been found in isolates from Belgium and Switzerland [34,133].

Among the *S.* Hadar isolates with partial or complete ciprofloxacin resistance, none showed mutations in GyrB. The ParC subunit in *S.* Hadar frequently carries the known Thr57Ser or Ser80Ile mutations (Appendix A). Furthermore, an isolate from Malaysia featured a novel double mutation in ParC, namely Arg96Ser and Pro98Lys [108], but the role of these mutations in ciprofloxacin resistance remains to be defined.

An earlier study reported that the position of the mutation in GyrA correlates strongly with the place of origin of an *S.* Hadar strain, with mutations at position 83 usually being found in isolates from Southeast Asia and mutations at position 87 being found in isolates from Southern Europe and North Africa [136]. However, in vitro induction of mutations in *S.* Hadar demonstrated that exposure of this serovar to ciprofloxacin usually first resulted in the acquisition of a mutation at position 87 [137], indicating that the geographic distribution of certain mutations may be the result of the use of particular quinolones in these regions. Our literature survey supports this notion.

Plasmid-encoded resistance determinants found in *S.* Hadar isolates with QRDR mutations include the *qnrD* gene [34,133]. An *S*. Hadar isolate collected in Switzerland was found to carry the novel *qnrD2* variant (Appendix A). This variant differs from *qnrD1* by two substitutions, Ile189Thr and Leu202Phe, but these have no effect on ciprofloxacin resistance [133]. Recently, an *S.* serovar Hadar isolate with an MIC of ciprofloxacin of 0.5 mg/L but having no QRDR mutations was found to carry the *qnrB19* gene [138]. Efflux of ciprofloxacin in *S.* Hadar isolates was only investigated in one study, which found that it did contribute to reduced ciprofloxacin susceptibility [106].

#### 4.2.4. *Salmonella* Kentucky (*S*. Kentucky)

*S.* Kentucky is one of the most frequently isolated serovars from poultry carcasses and the spread of the ciprofloxacin-resistant *S.* Kentucky ST198 isolate is posing a potential threat to farm animals (chicken, turkey and swine), wildlife and humans [139,140]. A research-based survey carried out in 28 countries [140] reported the international dissemination of the ST198 strain with isolates reported from the European Union [141] and North America [142,143].

The mutations in *S.* Kentucky isolate ST198 have already been comprehensively discussed by Le Hello et al. [140] and will therefore not be covered in this review. According to Le Hello et al., the particular substitution of GyrA residue Asp87 is characteristic for *S.* Kentucky ST198 isolates from different geographic regions, with North African and Southeast Asian isolates being characterized by Asp87Asn, Middle Eastern isolates by both Asp87Asn and Asp87Gly, East African and Indian isolates by Asp87Tyr, and West African isolates by Asp87Gly [140]. This literature survey found an *S.* Kentucky ST198 isolate from a patient in Switzerland carrying the Asp87Asn mutation. This patient likely contracted the pathogen during a trip to Libya, where this mutation is prevalent [144].

Our literature survey indicates the prevalence of ciprofloxacin-resistant *S.* Kentucky strains in Africa [30,41] Europe [28,34,133] and Asia [41,145] (Appendix A). Many of these strains are either confirmed or suspected to be *S.* Kentucky ST198 [28,30,41,144]. Analysis of the QRDR mutations reveals that *S.* Kentucky is prone to altering its chromosomal DNA in response to exposure to ciprofloxacin. Characteristic of ciprofloxacin-resistant isolates of *S.* Kentucky is the combination of a single mutation in GyrA, typically Ser83Phe, or a double mutation, Ser83Phe and substitution of Asp87 to Tyr, Gly or Asn, with single or double mutations in the ParC subunit of topoisomerase IV (Appendix A).

The contribution of ParC mutations to conferring ciprofloxacin resistance in *S.* Kentucky is debatable. A Thr57Ser mutation in ParC was identified in two ciprofloxacin-susceptible *Salmonella* control strains as well as in an *S.* Kentucky isolate susceptible to nalidixic acid in an earlier study [146], suggesting that this mutation is not involved in resistance [147] but rather represents an unimportant sequence variation [119]. The presence of double mutations in GyrA and ParC resulting in elevated resistance has been reported for an *S.* Kentucky ST198 epidemic clone [148]. In surveying *S.* Kentucky mutations, isolates from Belgium, all of them from humans, showed a double mutation in the *parC* gene, Thr57Ser and Ser80Ile, together with PMQR elements, namely *qnrB1* or *qnrD*. It should be noted that the clonal spread of the *S.* Kentucky ST198 genotype in Belgium was reflected in the PFGE-*Xba*I profiles of seven randomly chosen isolates, which showed 92% sequence identity [34]. Another notable feature of the *S.* Kentucky strain is that although mutation of Ser80 is common, it is always substituted by Ile and never by Arg (Appendix A).

Ciprofloxacin efflux has not been studied in most of the *S.* Kentucky isolates. However, mutations in the regulatory *ramR* gene have been reported to enhance AcrAB expression in some *S.* Kentucky ST198 isolates, thus increasing ciprofloxacin efflux [41]. Studies investigating the effect of mutations in RamR found a two- to six-fold increase in AcrA expression, accounting for the elevated resistance in these isolates [41,146].

Our literature survey found that *S.* Kentucky ST198 isolates expressing PMQR genes have exceptionally high MIC for ciprofloxacin, namely 32 and 24 mg/L for isolates expressing *qnrB1* and *qnrD*, respectively (Appendix A). It must be noted, however, that no PMQR genes were identified in an earlier study of *S.* Kentucky ST198 isolates [139].

#### 4.2.5. *Salmonella* Indiana (*S*. Indiana)

*S.* Indiana is a highly pathogenic strain that infects both humans and birds and causes acute enteritis and diarrhea. *S.* Indiana isolates have been identified in China, mainly in individuals working in meat retail, slaughter houses and chicken farms [128].

Our survey of QRDR mutations and their co-existence with PMQR genes revealed an alarming situation for *S.* serovar Indiana [149,150]. Most isolates carry both a double mutation in GyrA as well as a single or double mutation in the ParC subunit of topoisomerase IV. Substitution of GyrA residue Ser83 by Phe is prevalent with substitutions by Ile, Leu and Tyr being less frequent. In contrast, Asp87 is equally often substituted by Asn or Gly, but has not yet been found to be substituted by Tyr (Appendix A). Besides these most commonly mutated residues, analysis of QRDR mutations in *S.* Indiana isolates from China’s Shandong Province [150] and Henan Province [151,152] revealed additional mutations. *S.* Indiana isolates collected in Henan Province showed a prevalence of the Ser80Arg substitution in ParC [151,152] while isolates from Shandong Province carried the Ser80Ile substitution (Appendix A) [150]. *S.* Indiana isolates from Henan Province were found to have more transferable resistance elements. The Ser80Ile substitution and transferable resistance elements were also found in isolates collected from China’s Guangdong Province [149].

An in vitro study found that exposure of a parent *S.* Indiana strain, isolated from pig with no QRDR mutations and an MIC of ciprofloxacin of 0.125 mg/L, to ciprofloxacin at a concentration of 0.25 mg/L resulted in progeny with an MIC of 0.5 mg/L that acquired the Ser83Phe mutation in GyrA. An increase in the ciprofloxacin concentration to 1 mg/L resulted in progeny that retained the Ser83Phe mutation but displayed an increased MIC of 4 mg/mL [128]. A further increase in the ciprofloxacin concentration to 4 mg/L resulted in progeny with an MIC of 16 mg/L that acquired an additional GyrA mutation, Asp87Gly. Yet another increase in ciprofloxacin concentration to 8 mg/L resulted in progeny with an MIC of 32 mg/L that acquired an additional mutation in ParC, Thr57Ser [128]. In vivo *S.* Indiana isolates have not been investigated for efflux activity (Appendix A), but in vitro-selected ciprofloxacin-resistant mutants showed an increase in ciprofloxacin accumulation when treated with efflux pump inhibitor, indicating the presence of an active efflux system [128]. The contribution of the novel mutations in conferring FQs resistance thus remains to be defined. The role of a putatively novel Cys72 Gly mutation in ParC [153] also remains unclear, because our sequence alignment indicates that Gly is actually the wild-type residue at this position in *S*. Typhimurium str. LT2 (Figure 6).

*S.* Indiana is the only serovar to date that was found to carry all the known PMQR determinants [154], except for *qnrC*. Among these PMQR determinants, the *aac(6′)-Ib-cr* gene is the most prevalent in *S.* Indiana, followed by the *oqxAB*, and the *qepA, qnrA, qnrB, qnrD* and *qnrA* genes, which are much less prevalent [154]. Given the universal occurrence of transferrable resistance determinants in *S.* Indiana, their dissemination through conjugation poses a serious threat. However, under laboratory conditions, conjugation was usually unsuccessful, with the exceptions of studies by Bai et al. and Jiang et al., whichfound transconjugants carrying the *oqxA*, *oqxB* and *aac*(*6′*)-*Ib-cr* genes [151] and the *qnrA, aac(6)-Ib-cr* and *oqxA* genes, respectively [149].

#### 4.2.6. *Salmonella* Infantis (*S*. Infantis)

*S.* Infantis is another poultry-related *Salmonella* serovar. Based on the analysis of isolates from Europe [28,34,155], Asia [114] and South America [32], *S.* Infantis isolates have a high prevalence of the Ser83Tyr mutation in GyrA together with the Thr57Ser mutation in ParC (Appendix A). Exceptions include isolates from Iran, which were found to mostly carry a mutation of Asp87 to Tyr or Asn [91,156]. In an *S.* Infantis isolate from Iran, the novel Leu41Pro mutation was found to co-exist with the Asp87Tyr mutation [91] (Appendix A). However, it is not yet clear whether this variation represents a sequence polymorphism or an actual ciprofloxacin-resistance determinant.

To date, only a single GyrB mutation has been found in *S.* Infantis, the novel Leu470Met substitution. Because this mutation co-exists with GyrA mutation Ser83Tyr and ParC mutation Thr57Ser, it remains to be determined whether it actually contributes to the reduced ciprofloxacin susceptibility of this isolate [28]. The only PMQR determinant reported to date for *S.* Infantis isolates with QRDR mutations is *qnrA1* (Appendix A).

#### 4.2.7. *Salmonella* Derby (*S*. Derby)

*S.* Derby is one of the most prevalent serovars in pigs and pork-related products in Europe, North America and Asia, ranking among the top 10 human-associated *Salmonella* serovars [153,157]. *S.* Derby isolates identified in Europe were found to be susceptible to ciprofloxacin [157,158,159].

In our literature survey, all isolates from Europe [34] and Asia [134,153] were found to have a single mutation in GyrA (Appendix A). However, isolates from Europe and Asia differ in that isolates from Europe also carry a ParC mutation but no PMQR genes [34], whereas the opposite is true for isolates from Asia [134,153].

In a comprehensive study, Lin et al. found that pork samples collected from retail meat in China harbored *S.* Derby with very high ciprofloxacin resistance reflected in their MIC of 2 mg/L [153]. The isolates featured a novel GyrA mutation, Asn78His, but since they also expressed the acetyl transferase enzyme, the contribution of this new mutation to ciprofloxacin resistance remains to be determined. It is worth mentioning that our current survey finds that His is in fact the wild-type residue at position 78 in *S*. Typhimurium str. LT2 (Figure 3) and not Asn, as stated in this study [153]. Many *S.* Derby isolates having a single target gene mutation were found to carry three PMQR genes (Appendix A) [153]. The presence of these genes in such a high number is alarming, but in a study of Chinese isolates, none of these ciprofloxacin-resistance determinants could be transferred to a recipient *E. coli* strain. Analysis of a representative *S.* Derby isolate revealed that the *aac(6′)-Ib-cr*, *oqxAB* and *qnrS2* genes were all integrated in the chromosomal DNA, rather than being carried on a plasmid, but the *aac(6′)-Ib-cr* gene was also detected on a plasmid [153]. The integration of the *oqxAB* gene into the chromosomal DNA, which has already been reported in an earlier study, suggests clonal dissemination of *oqxAB*-positive *S.* Derby in the local food supply network of Hong Kong [160].

An *S.* Derby isolate with an MIC of ciprofloxacin of 2 mg/L but havng no QRDR mutations was found to carry four PMQR genes, namely *aac(6′)-Ib-cr*, *oqxAB*, *qnrS2* and *qnrB8*, as well as to have the *oqxAB* and *qnrS2* genes integrated in its chromosome [153]. The efflux efficiency of chromosomally integrated *oqxAB* genes is unclear, because the inhibition of the efflux system in this isolate and the one described above with Phe-Arg-β-naphthylamide did not result in a substantial decrease in the MIC, indicating the absence of an efficient efflux activity. However, the MIC of another isolate with no QRDR mutations and chromosomally integrated *oqxAB*, *qnrS2* and *aac(6′)-Ib-cr* dropped from 16 mg/L to 2 mg/L in the presence of an efflux pump inhibitor [153].

#### 4.2.8. *Salmonella* Newport (*S*. Newport)

*S.* Newport has been isolated from human clinical samples, retail meat and seafood. Analysis of mutations in the QRDRs of *S.* Newport isolates from Europe [28,34], Asia [108,134] and South America [112] showed that double mutations are prevalent with one mutation in the GyrA subunit, either at position 83 or 87, and another mutation in the ParC subunit, usually Thr57Ser (Appendix A). A Malaysian isolate had only a single mutation in GyrA but a novel double mutation in ParC, Arg96Ser and Pro98Thr [108].

A study from India reported that exposure of an originally susceptible *S.* Newport isolate to increasing concentrations of nalidixic acid resulted in mutant progeny with an MIC of >256 mg/L carrying the Asp87Gly mutation and displaying active efflux activity [161]. Simultaneous screening of in vivo nalidixic acid-resistant *S.* Newport isolates found that isolates carrying the Asp87Asn mutation in GyrA have an MIC of nalidixic acid of >256 mg/L [161]. This study thus showed a prevalence of the Asp87 mutation in nalidixic acid-resistant *S.* Newport isolates. However, a study by Wasyl et al. reported the prevalence of not only Asp87Gly in *S.* Newport isolates but also of the Ser83Tyr and Ser83Phe mutations (Appendix A) and diverse PFGE-*Xba*I profiles, hinting at the contribution of several clones to the spread of *S.* Newport [28], a notion that is consistent with our literature survey.

In terms of PMQR genes, *qnrS1 and qnrS3* have been found in *S.* Newport isolates from Europe [28,34] and Asia [108]. Isolates from turkey meat in Denmark with MICs ranging from 0.5 to 1 mg/L carried a different PMQR determinant, *qnrB5*, as well as the Thr57Ser mutation in ParC [162]. *S.* Newport strains with a non-classical phenotype that escape routine susceptibility assays pose a serious threat [107]. Moreover, plasmid-borne ciprofloxacin resistance is more common in this serovar than chromosomally encoded resistance [163].

#### 4.2.9. *Salmonella* Virchow (*S*. Virchow)

*Salmonella* serotype Virchow ranks among the five most abundant *Salmonella* serovars in Europe [164] and is mainly linked to imported poultry meat. A study from Portugal found a high level of quinolone resistance in *S*. Virchow isolates [165].

In a study of *S*. Virchow isolates collected in Switzerland, all of them carried the same Asp87Tyr substitution in GyrA, including isolates associated with travel to Thailand, suggesting a well-established *S*. Virchow genotype with quinolone resistance due to this target gene mutation [133]. However, a simultaneous report by Wasyl et al., studying quinolone-resistance mechanisms among *Salmonella* isolates from Poland, identified diverse GyrA mutations (Appendix A) in three *S.* Virchow isolates with indistinguishable PFGE-*Xba*I profiles, including a double mutation in GyrA with intermediate ciprofloxacin susceptibility [28]. Overall, our literature survey establishes that there is no preferred site of mutation in GyrA associated with quinolone resistance in *S.* Virchow isolates, but among the most frequent mutations are the Ser83Phe and Asp87Tyr substitutions.

Mutations in GyrB, ParC and ParE have not been identified in any *S.* Virchow isolate from Europe [28,34,133], Africa [30] and Asia [119,134,166] (Appendix A). One isolate from Thailand was found to have a Val95Leu mutation in ParC, but this mutation was also present in ciprofloxacin-susceptible isolates and was therefore excluded from playing a role in ciprofloxacin resistance [119]. Based on our sequence alignment study, the wild-type residue at position 95 in ParC is Tyr (Figure 6). PMQR genes identified in *S.* Virchow isolates include *qnrS1* and *qnrB2* genes [133,166]. Conjugational transfer of the PMQR genes was not successful for all *S.* Virchow isolates, except for one Swiss isolate bearing a plasmid with the *qnrS1* gene. An *S.* Virchow isolate collected from retail pork in China carried four PMQR genes, *aac-(6′)-Ib-cr, oqxAB, qnrS8* and *qnrD*. This isolate showed a high resistance to ciprofloxacin with an MIC of 4 mg/L, even though it carried no QRDR mutations [153]. The mutations identified in type-II topoisomerases of NTS have been depicted in Figure 4, Figure 5, Figure 6 and Figure 7 for GyrA, GyrB, ParC and ParE, respectively. The summary of these mutations is also given in Figure 8.

## 5. Conclusions

This literature review illustrates how *Salmonella* serovars evade the consistent evolutionary pressure associated with the use of FQs to treat typhoidal and non-typhoidal salmonellosis by acquiring mutations in their QRDRs, as well as PMQR determinants and/or efflux mechanisms. Typhoidal *Salmonella*, in particular *S.* Typhi and *S.* Paratyphi A, respond to antibiotics exposure predominantly by acquiring the Ser83Phe substitution in GyrA. While Ser83 mutations have previously been reported to account for more than >90% of all QRDR mutations [16,167], the acquisition of the *qnrS1* gene is another mechanism by which *S.* Typhi adapts to ciprofloxacin pressure. *S.* Enteritidis and *S.* Typhimurium are the most extensively studied non-typhoidal serovars. The Asp87Tyr substitution is most prevalent in ciprofloxacin-resistant *S.* Enteritidis isolates, but the Ser83Phe substitution is also common. *S.* Typhimurium isolates show even less prevalence for a specific GyrA mutation in response to ciprofloxacin exposure, with Ser83 and Asp87 mutations having very similar frequencies. Ciprofloxacin extrusion mediated by efflux transporters is another resistance mechanism that cannot be ruled out for *Salmonella* serovars Typhimurium and Enteritidis. Our literature review establishes *S.* Kentucky and *S.* Indiana as emerging threats to food imports, because increasing numbers of isolates are found to be ciprofloxacin resistant, and also emphasizes the critical importance of surveilling *S.* Typhimurium, *S.* Indiana and *S.* Derby to control the spread of transferable ciprofloxacin-resistance determinants.

Several issues require further attention. An important question is whether the complete resistance of many of the single-mutation Salmonella strains is indeed just due to the acquisition of the single QRDR mutations or is in fact also based on contributions by other resistance mechanisms, such as reduced porin expression or increased transporter-mediated efflux. While several new mutations have been identified in currently circulating typhoidal and non-typhoidal *Salmonella* strains, more studies are needed to understand their role in ciprofloxacin resistance. In this respect, our literature survey highlights the importance of careful analysis of the correct reference nucleotide sequence to identify novel mutations. The emergence of non-classical resistant *Salmonella* strains, characterized by high susceptibility to nalidixic acid and reduced or intermediary susceptibility to ciprofloxacin, is a major threat for the dissemination of drug-resistant *Salmonella* serovars. Because routine screening uses nalidixic acid to evaluate the drug resistance of Salmonella serovars, non-classical *Salmonella* strains are not detected, establishing the need for improved diagnostic procedures. The substantial role that transporter-mediated efflux plays in quinolone resistance poses a need for further investigations, particularly for natural isolates. The sparsity of data for Southeast Asia also calls for research-based surveys in this area, where *Salmonella* is endemic. Finally, the lack of studies focusing on less frequent serovars, such as *S.* Choleraesuis, *S.* Dublin, *S.* Heidelberg, *S.* Rissen, *S.* Schwarzengrund, *S.* Stanley, *S.* Tennessee and others, makes it impossible at this point to establish the presence of mutations potentially conferring antibiotic resistance and thus to assess the risk they pose to humanity.

In conclusion, this overview of QRDR mutations in *Salmonella* serovars provides a snapshot of currently circulating ciprofloxacin-resistant strains, but *Salmonellae* have also developed resistance to other antibiotics. Azithromycin is among the non-FQ antibiotics being used for the treatment of salmonellosis, particularly typhoidal salmonellosis. However, azithromycin has already been reported against and the molecular mechanism underlying this resistance phenotype has been identified as mutations in the inner membrane efflux protein AcrB [168] and its regulatory protein AcrR [169]. Antibiotics belonging to the cephalosporin family have also been used to treat typhoid fever. Again, resistance to these antibiotics has already emerged in *Salmonella* and in this case is mediated by plasmid-borne factors [170]. Typhoidal *Salmonella* are endemic in developing countries and food imports are the cause for the occurrence of non-typhoidal *Salmonella* in developed countries. A coordinated effort between developed and developing nations is of critical importance to curtail the further spread of *Salmonella* among communities worldwide. Moreover, there is urgent need to fight drug resistance to combat the emergence of extensively multi- and pan-drug-resistant *Salmonella* strains. Currently, different plant-based inhibitors and synthetic peptides are being investigated for their capacity to fight drug resistance [171,172].

## Figures and Tables

**Figure 1 antibiotics-10-01455-f001:**
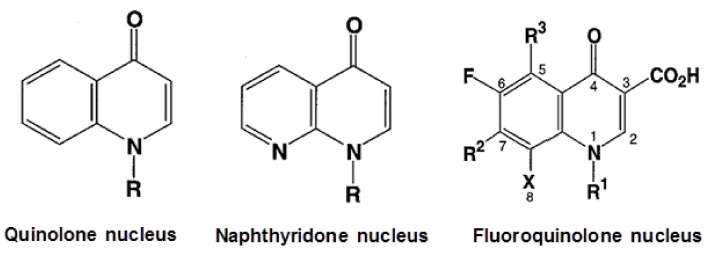
Structures of the quinolone, naphthyridone, and fluoroquinolone nuclei. ‘R’ denotes an alkyl group, ‘X’ denotes a small substituent, such as F, Cl, or CH3O^−^.

**Figure 2 antibiotics-10-01455-f002:**
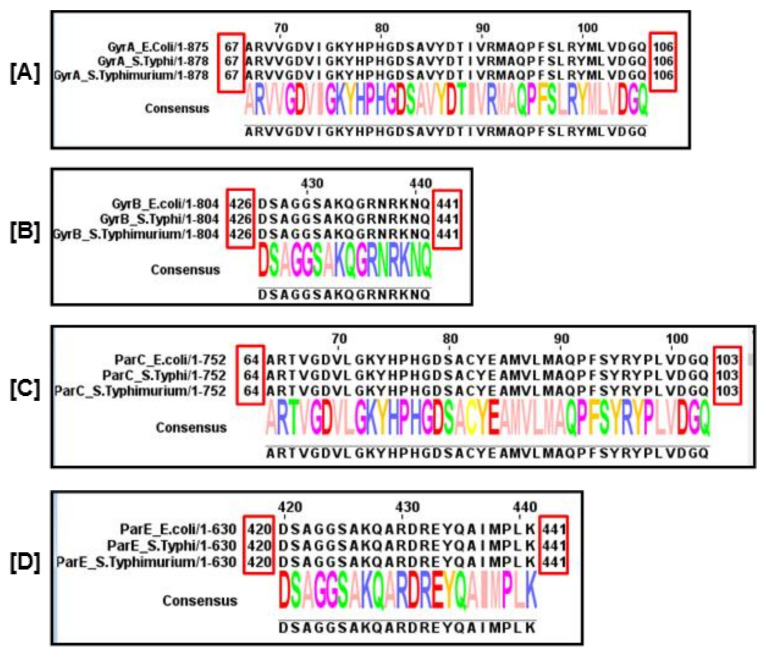
Amino acid sequence alignment for the QRDR of *E. coli*, *S*. Typhi and *S*. Typhimurium homologs. **(A)**; GyrA QRDR (Ala67-Gln106) (**B**); GyrB QRDR (Asp426-Lys447) (**C**); ParC QRDR (Ala64 to Gln103) (**D**); ParE QRDR (Asp420-Lys441). The alignment was performed with MUSCLE using Jalview [23]. The complete sequence alignment of all these genes is shown in Appendix A for GyrA, GyrB, ParC, and ParE, respectively.

**Figure 3 antibiotics-10-01455-f003:**
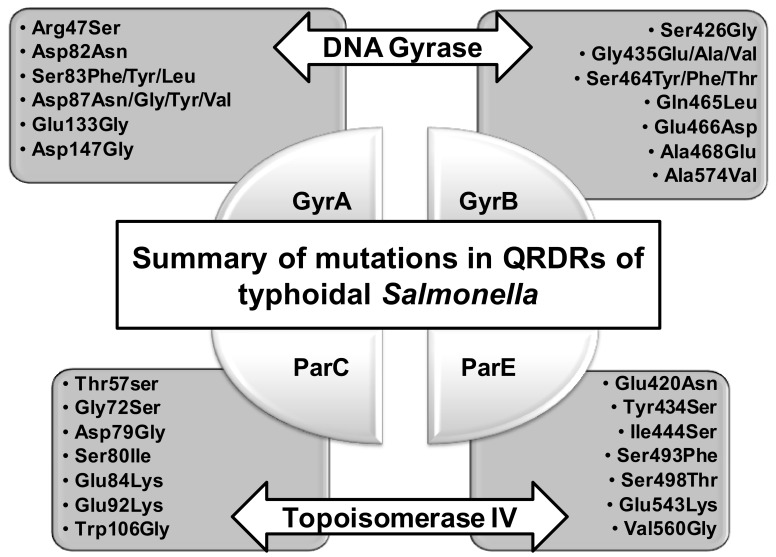
Mutations found in the QRDRs of DNA gyrase (GyrA and GyrB) and topoisomerase IV (ParC and ParE) of typhoidal *Salmonella*.

**Figure 4 antibiotics-10-01455-f004:**
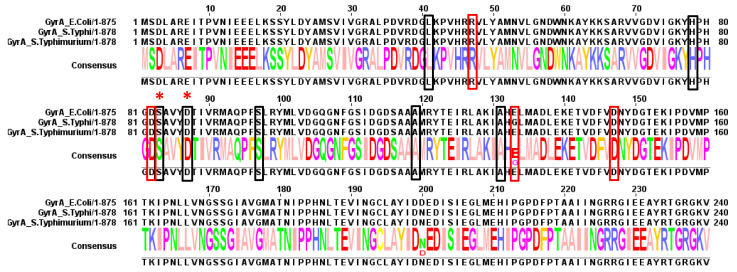
Summary of mutations found in GyrA. Red boxes indicate residues found to be mutated in typhoidal *Salmonella*. Black boxes indicate residues found to be mutated in non-typhoidal *Salmonella* (discussed below) and “*” indicate residues found to be mutated in both.

**Figure 5 antibiotics-10-01455-f005:**
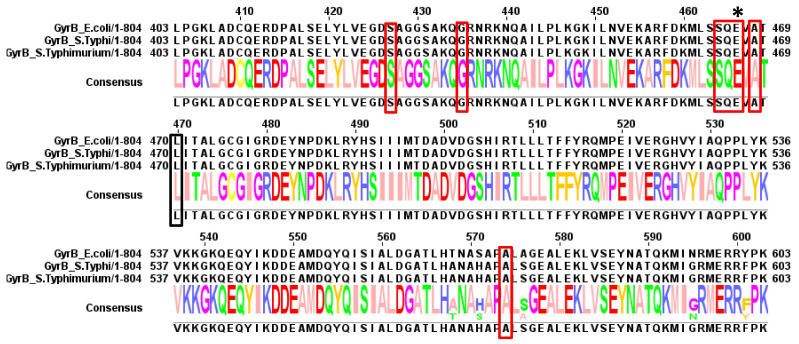
Summary of mutations found in GyrB. Red boxes indicate residues found to be mutated in typhoidal *Salmonella*. Black boxes indicate residues found to be mutated in non-typhoidal *Salmonella* (discussed below) and “*” indicate residues found to be mutated in both.

**Figure 6 antibiotics-10-01455-f006:**
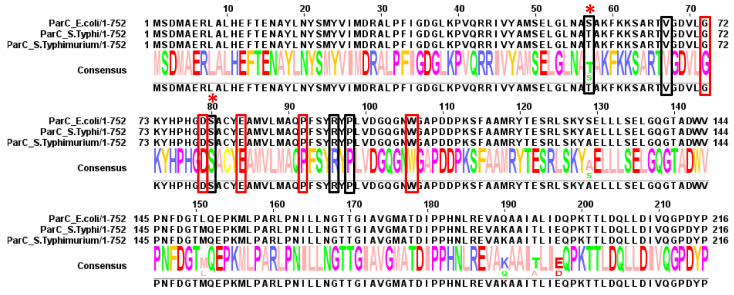
Summary of mutations found in ParC. Red boxes indicate residues found to be mutated in typhoidal *Salmonella*. Black boxes indicate residues found to be mutated in non-typhoidal *Salmonella* (discussed below) and “*” indicate residues found to be mutated in both.

**Figure 7 antibiotics-10-01455-f007:**
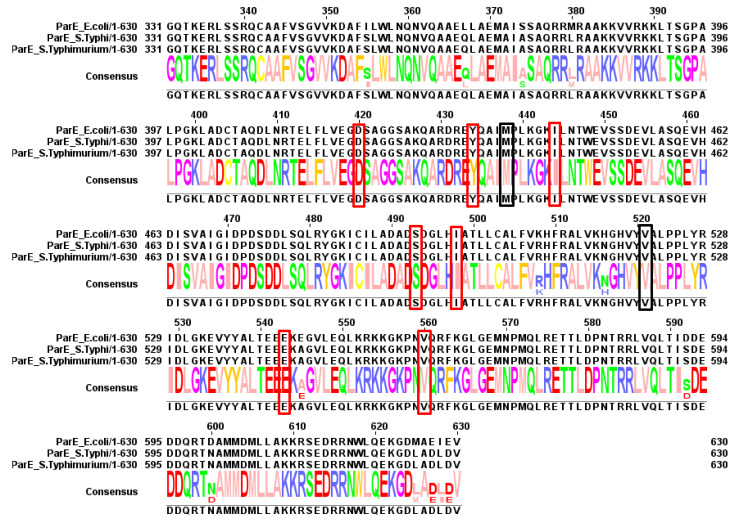
Summary of mutations found in ParE. Red boxes indicate residues found to be mutated in typhoidal *Salmonella*. Black boxes indicate residues found to be mutated in non-typhoidal *Salmonella* (discussed below).

**Figure 8 antibiotics-10-01455-f008:**
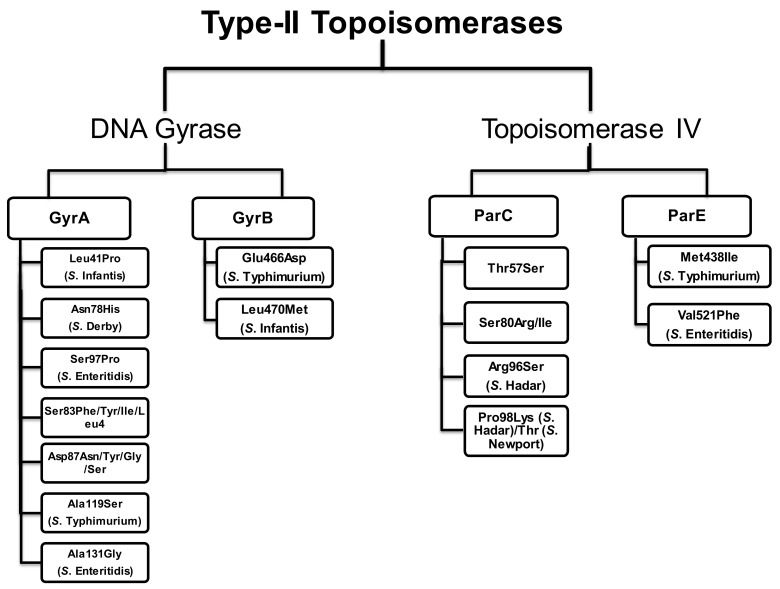
Mutations in the QRDRs of NTS.

**Table 1 antibiotics-10-01455-t001:** Genbank accession numbers of type-II topoisomerases used for sequence alignment.

Bacteria	GyrA	GyrB	ParC	ParE
	Genbank Accession Numbers			
*E. coli* O157:H7 str. EDL933	AAG57360.1	AAG58896.1	AAG58155.1	AAG58169.1
*S.* Typhi str. Ty2	AAO68297.1	AAO71180.1	AAO70639.1	AAO70645.1
*S.* Typhimurium str. LT2	AAL21173.1	AAL22694.1	AAL22048.1	AAL22055.1

**Table 2 antibiotics-10-01455-t002:** List of *Salmonella* serovars analyzed for mutations in their QRDRs.

	*Salmonella enterica* subsp. *enterica*	Discussed In
Typhoidal *Salmonella*	Typhi	Section 4.1
	Paratyphi A, B and C	Section 4.1
Non-typhoidal *Salmonella* (NTS)	Enteritidis	Section 4.2.1
	Typhimurium	Section 4.2.2
	Hadar	Section 4.2.3
	Kentucky	Section 4.2.4
	Indiana	Section 4.2.5
	Infantis	Section 4.2.6
	Derby	Section 4.2.7
	Newport	Section 4.2.8
	Virchow	Section 4.2.9

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
