# Peer review of "Mutational Diversity in the Quinolone Resistance-Determining Regions of Type-II Topoisomerases of Salmonella Serovars"

_antibiotics, 2021, doi:10.3390/antibiotics10121455_

Round 1

Reviewer 1 Report

This review is intended to provide the following information (membrane trans-31 Porter-mediated quinolone efflux on quinolone resistance in strains with QRDR mutations, information on routine surveillance of food-borne Salmonella serotypes, particularly to improve laboratory diagnosis of quinolone-resistant Salmonella strains. )

However, these papers are currently being studied in various ways. Similar reviews are also provided, so the differences between these reviews are not noticeable.

Aside from listing information on strains, no outstanding comprehensive analysis is presented.

Above all, it does not present a more advanced idea of ​​drug-resistant Salmonella used for diagnosis.

It is expected that this paper will present ideas for advanced analysis and utilization based on information on drug resistance.

Author Response

Regards,

Reviewer 2 Report

The authors give a comprehensive review related to the mutational diversity in quinolone resistance-determining regions (QRDRs) of Salmonella serovars. The review is composed of several parts that comprise (1) Introduction, (2) Mode of action of fluoroquinolones and QRDRs, (3) Known mechanisms of quinolone resistance, (4) Mutations in the QRDRs of Salmonella serovar, and (5) Conclusions. A predominant part of the body text is dedicated to section 4 (Mutations in the QRDRs of Salmonella).

I found only minor points for correction:

  1. The title needs improvement to avoid fragmentation in separate parts. Semicolon does not represent a good writing style.
  2. It will be helpful for the reader to have a table that lists Salmonellas' names from section 4. I counted around 10 different variations of Salmonellas.
  3. Not much attention was paid to fighting against bacterial resistance. The authors could include a short discussion related to the importance of synthetic peptides to fight resistance. Relevant references could be: (1) Prediction of amphiphilic cell-penetrating peptide building blocks from protein-derived amino acid sequences for engineering of drug delivery nanoassemblies, The Journal of Physical Chemistry B, 2020, 124, 4069-4078., and (2) Deep Learning Enables Discovery of a Short Nuclear Targeting Peptide for Efficient Delivery of Antisense Oligomers, JACS Au 2021, doi.org/10.1021/jacsau.1c00327

Author Response

Regards,
